# Study on Toughening and Temperature Sensitivity of Polyurethane Cement (PUC)

**DOI:** 10.3390/ma15124318

**Published:** 2022-06-18

**Authors:** Ning Hou, Jin Li, Xiang Li, Yongshu Cui, Dalu Xiong, Xinzhuang Cui

**Affiliations:** 1School of Transportation Civil Engineering, Shandong Jiaotong University, Jinan 250357, China; 20107043@stu.sdjtu.edu.cn (N.H.); 20107012@stu.sdjtu.edu.cn (Y.C.); 2Shandong Hi-Speed Construction Management Group Co., Ltd., Jinan 250001, China; sdyslixiang@163.com; 3Jinan Kingyue Highway Engineering Company Limited, Jinan 250220, China; glghgl@126.com; 4School of Civil Engineering, Shandong University, Jinan 250061, China; cuixz@sdu.edu.cn

**Keywords:** polyurethane cement (PUC), PVA fiber, carbon fiber, steel fiber, temperature sensitivity, BP neural network

## Abstract

Polyurethane cement (PUC) is now commonly used in the reinforcement of old bridges, which exhibit various issues such as poor toughness, temperature-sensitive mechanical properties, and brittle failure. These problems can lead to the failure of the reinforcement effect of the PUC on old bridges in certain operating environments, leading to the collapse of such reinforced bridges. In order to alleviate these shortcomings, in this study, the toughness of PUC is improved by adding polyvinyl alcohol (PVA) fiber, carbon fiber, and steel fiber. In addition, we study the change law of the flexural strength of PUC between −40 °C and +40 °C. The control parameters evaluated are fiber type, fiber volume ratio, and temperature. A series of flexural tests and scanning electron microscope (SEM) test results show that the flexural strength first increases and then decreases with the increase in the volume-doping ratio of the three fibers. The optimum volume-mixing ratios of polyvinyl alcohol (PVA) fiber, carbon fiber, and steel fiber are 0.3%, 0.04% and 1%, respectively. Excessive addition of fiber will affect the operability and will adversely affect the mechanical properties. The flexural strength of both fiber-reinforced and control samples decreases with increasing temperature. Using the flexural test results, a two-factor (fiber content, temperature) BP neural network flexural strength prediction model is established. It is verified that the model is effective and accurate, and the experimental value and the predicted value are in good agreement.

## 1. Introduction

Polyurethane cement (PUC) made of polyurethane and cement as the main materials has a series of advantages such as good durability, high strength, and good bonding performance. PUC has been applied to reinforce old bridges such as T-shaped and hollow slab bridges [1,2,3,4,5]. Many scholars have carried out in-depth studies on PUC. Haleem K., Guiwei Liu et al. [6,7,8,9], studied the basic mechanical properties of PUC and conducted flexural strengthening tests on seven T-beams with different degrees of damage, and the analysis concluded that PUC can significantly improve the bridge deformation and load-bearing capacity. PUC has high brittleness and its failure form is brittle failure. Kexin Zhang et al. [10] embedded a prestressed steel wire rope in PUC to strengthen a reinforced concrete T-beam, which significantly increased the ultimate load and stiffness of the T-beam, but did not change the brittleness of the PUC itself. Yasin Onuralp Özkılıç et al. [11] found that adding steel fibers can change concrete’s failure form from brittle failure to ductile failure. Ralph Jabbour et al. [12] found that adding polypropylene fibers, polyvinyl alcohol (PVA) fibers and steel fibers to concrete can improve the deflection and ductility of concrete, and the overall effect of adding polyvinyl alcohol fibers achieves the best results. Yang Nan et al. [13] conducted a study to improve the mechanical properties of PUC by the method of carbon fiber reinforcement, and the study showed that the amount of carbon fiber reinforcement is the main factor affecting the splitting and flexural strength of polyurethane cement composites. Therefore, the addition of fibers to PUC to improve toughness and flexural strength has feasibility and research value.

In addition, the latitude span between the north and south of China is large. The lowest temperature in the winter reaches about −40 °C in cold areas in the north, and the highest temperature in the summer is close to +40 °C in hot areas. Hongshuai Gao et al. [14] found that temperature has a great influence on the fatigue life of PUC in their study in which PUC was used in a steel bridge deck pavement. PUC shows large strength differences under different ambient temperatures in practical applications, which in turn affect the bridge-strengthening effect. Thus, it is necessary to investigate the law of the influence of temperature on the mechanical properties of PUC.

The application of artificial intelligence methods to the design and application of materials has become a current research hotspot. It has good prospects in the design of building material mix ratios. Duam et al. [15] used an artificial neural network to establish a model for predicting the elastic modulus of recycled concrete suitable for different recycled aggregate sources. Zhang and others [16] used a BP neural network method to study the influence of the water–binder ratio and polycarboxylate superplasticizer content on the mechanical properties of ultra-high-performance concrete. Artificial neural networks imitate the behavioral characteristics of animal neural networks and can solve highly nonlinear complex system problems. They have been widely used in prediction problems in various fields such as logistics, transportation, medicine, and civil engineering [17,18,19,20]. Finding available mix ratios for materials often requires extensive, repetitive testing, which is a time-consuming and expensive process. Applying a BP neural network to the research of fiber-reinforced PUC can save time and costs related to subsequent research.

In this work, this research selected three kinds of toughening fibers, namely, PVA fiber, carbon fiber, and steel fiber, and this research explored the optimal toughening volume-mixing ratio of PUC by adjusting the fiber content to observe the change in material strength. The effect of three kinds of fibers on the flexural strength of PUC under different fiber-mixing ratios and different temperatures was examined, and the toughening mechanism of various fibers and the reasons for the influence of temperature were analyzed. According to the existing experimental data, this research designed a BP neural network flexural strength prediction model with the fiber volume ratio and temperature as the input layer, and verified that the model can predict well with suitable accuracy. It provides a reference for the future use of PUC for bridge reinforcement designs in different regions.

## 2. Materials and Methods

### 2.1. Materials

#### 2.1.1. PUC Composite

The polyurethane cement (PUC) composite used in this study was composed of isocyanate, modified polyether, defoaming agent, cement, and other components. Isocyanate (purity of ≥99%, density of 1.23 g/cm^3^), modified polyether (HYPOP-3628, hydroxyl value of 25–29 mgKOH/g, moisture of ≤0.08%, viscosity of 2200–3000 mPa·s at 25 °C), and defoaming agent (Shandong Wanhua Chemical Group, Zibo, China), when modified by the manufacturer, can slow down the chemical reaction rate during curing. Portland cement labelled 42.5 was also used in this study. The composition of the PUC material is shown in Table 1.

#### 2.1.2. Fiber

Carbon fiber has high strength and a strong anti-aging ability; polyvinyl alcohol (PVA) fiber has low density; and steel fiber has high tensile and compressive strength. In this study, the above three fibers with different performance characteristics were selected for the preparation of fiber-strengthened PUC, and the reinforcing effect of each fiber was compared and analyzed. The fiber technical indexes are shown in Table 2, and fiber appearance is shown in Figure 1.

### 2.2. Test Block Preparation and Mix Ratio

According to the results of the fiber volume mixing ratio test in the previous attempt of this study, a suitable volume-mixing ratio range for each fiber was selected for testing (higher ratios are not feasible because they are difficult to mix well, which will be discussed later). See Table 3 for the specific mix ratio and coding.

This research weighed the materials according to the mixing ratio, and dry-mixed the cement and fibers for 2 min to ensure that the fibers were evenly distributed. Then, this research added isocyanate and stirred for 1 min. Finally, modified polyether and defoaming agent were added and stirred rapidly for 3 min. According to the DLT 5126-2001 Test Rules for Polymer-Modified Cement Mortar, the stirred PUC was then poured into the triple test mold (40 mm × 40 mm × 160 mm) with the release agent applied to the inner wall. It was then for 7 days under a consistent temperature at (20 ± 3) °C and a relative humidity above 60%. The test temperature was selected according to the temperature changes in summer and winter in various regions of China. The specimens were placed in different temperature variables (−40 °C, −20 °C, 0 °C, +20 °C, and +40 °C) for 2 h; then, the flexural test was performed.

### 2.3. Test Method

According to the DLT 5126-2001 Test Rules for Polymer-Modified Cement Mortar, the flexural strength and ultimate tensile strain of the beam bottom of the specimen were tested by the three-point bending test, and the improvement effect of fiber on the ductility of PUC was explored. The flexural strength of the specimen is calculated according to Formula 1. The average value of the flexural strength of the three test blocks was used as the flexural strength test result for each group of tests. The loading process is shown in Figure 2.
(1)$$\sigma =\frac{1.5PL}{b^{3}}$$
where *σ* is the flexural strength of the specimen (MPa), *P* is the applied load (N), *L* is the spacing between two supports (mm), and *b* is the width of the prismatic square section (mm).

## 3. Experimental Results and Analysis

### 3.1. Flexural Strength—Fiber Volume-Mixing Ratio Curve

As can be seen from Figure 3, with the increase in the volume-doping ratio of various fibers, the flexural strength of PUC at each temperature showed a trend of first increasing and then decreasing. The flexural strength of the same fiber at different temperatures showed essentially the same trend with the volume-doping ratio. The flexural strength–volume-doping ratio curves of PF-PUC, CF-PUC, and SF-PUC at a temperature of +20 °C were specifically analyzed (the test block number is shown in Table 3). The flexural strength of A1–A3 increased by 10.4%, 54.5%, and 16.8%; the flexural strength of B1–B3 increased by 19.2%, 31.4%, and 26.4%; and the flexural strength of C1–C3 increased by 11.4%, 32.8%, and 4.9%, respectively. The reason for the increased flexural strength is that the fibers are randomly and uniformly distributed to form a tightly connected network with the PUC. The integrity of the PUC is greatly improved, and the tension zone requires greater tension to pull the fibers out of the PUC or break them. The incorporation of fibers also corresponds to an increase in the content of hard segments in the polyurethane chain segments, thus improving the mechanical properties of the polyurethane cement composites.

At a temperature of +20 °C, the flexural strength of A4 and A5 test blocks decreased by 20.2% and 39.9%, and the flexural strength of B4 and B5 test blocks decreased by 18.4% and 38.8%; this is because when the fibers are over-doped, the fibers cannot be uniformly dispersed inside the test blocks and form agglomerates, resulting in PUC matrix defects within the interface. When subjected to external forces, the region becomes a weak point of stress and the energy consumed by the PUC matrix within the interface decreases, causing local damage to the matrix, resulting in a decrease in the flexural strength of PUC. Thus, PVA fibers, carbon fibers, and steel fibers have the best doping ratios of 0.3%, 0.04%, and 1.0%. When the fiber volume-doping ratio exceeds the best ratio, this will result in a greater loss of PUC flexural strength. The addition of PVA fiber to enhance the PUC flexural strength is one of the best effects.

### 3.2. Fiber-Toughening Mechanism and Specimen Failure Mode

The flexural strength of PUC is determined by the strength of polyurethane, the bond strength of the interface between polyurethane and cement, and the bond strength between polyurethane and fiber. After adding the three kinds of fibers, under the action of load, the stress is transmitted to the fibers through the PUC matrix, and the fibers can resist the local strain of the nearby matrix and bear a certain stress. When microcracks appear in the PUC matrix, the three fibers with high ductility straddle the microcracks to relieve sharp stress concentrations. The fibers absorb a large amount of destructive energy, so the PUC toughness is significantly improved. After flexural failure, each type of fiber has three failure modes, namely, being pulled out, sliding relative to the PUC matrix, and breaking.

Figure 4 shows the typical failure form of the specimen. The failure surface of the specimen without fiber is flat, and the crack develops vertically from the middle of the bottom of the beam to the top surface of the specimen. The PF-PUC, CF-PUC, and SF-PUC specimens were cracked from the middle of the bottom, then, the cracks extended along the inclined plane to the top of the specimens, and the failure sections of the specimens were irregular and uneven. This is because the mechanical riveting effect occurs between the fibers and the PUC matrix, and the fibers in the tension zone are effectively “bridging cracks”. When the fibers spanning both sides of the crack yield or break, energy is consumed, which requires more pull-out work and debonding work, and improves the deformation ability of the PUC. In Figure 4b–d, the inclination angle of the failure section of the specimen shows that the PUC after adding fibers is not destroyed instantaneously when cracks appear. On the contrary, after the crack appears, the fibers in the interface are stressed and the stress concentration is relieved. Then, the specimen enters the stage of bearing external force with cracks. Even if this time is maintained for a very short period, it provides more possibilities for the direction of crack development and increases the toughness of PUC.

The addition of fibers can significantly improve the ductility of PUC. Here, this research performed a comparative analysis of the ultimate micro-strain under the optimal dosage of various fibers at room temperature of +20 °C (see Figure 5). The limit micro-strain of PF-PUC and CF-PUC is close, and these two kinds of finer-diameter fibers will inhibit the development of deformation and fully play their role when the PUC matrix is subjected to micro-deformation. The ultimate elongation of PVA fiber is relatively large, so the PVA fiber bridging the crack is not easily prematurely broken and pulled out, and the fiber working time is longer, so the flexural strength of PF-PUC is improved the most, up to 58.5 MPa. The ultimate micro-strain of SF-PUC incorporating steel fibers with a larger diameter increased by 166.7%, indicating that the steel fibers can fully resist deformation when the PUC matrix is greatly deformed.

### 3.3. Flexural Strength–Temperature Curve

It can be seen from Figure 6 that when the temperature is −40 °C–+40 °C, with the increase in temperature, the flexural strength curves of PUC under various fiber-mixing ratios show a downward trend. The flexural strength advantage of PUC is more prominent in areas with lower temperature. The flexural strength of PUC with different volume-mixing ratios of the same fiber is roughly the same with temperature, and the flexural strength decreases by about 30% from −40 °C to +40 °C on average. It can be seen that the temperature has a considerable influence on the flexural strength of PUC, and PUC shows temperature sensitivity. The reduction rate of the flexural strength of the three kinds of fibers was slightly smaller than that of the control group, and the contribution of adding fibers to alleviate the change in the flexural strength of PUC caused by temperature difference was small. The flexural strength of the control group still reached 33.21 MPa when the temperature was 40 °C, reflecting the high-strength characteristics of PUC.

### 3.4. PUC Temperature Sensitivity Analysis

It can be inferred from Section 2.3 that the flexural strengths of the four groups of specimens all show a law of decreasing with the increase in temperature when the temperature is −40 °C–+40 °C. This is determined by the characteristics of the polyurethane material, and the fiber type and content have little influence on it.

Polyurethane is composed of rigid hard segments and flexible nested soft segments [21]. The hydrogen bonding between the molecular chains of the hard segment is the main driving force of the micro-phase separation, and has an important influence on the crystallization of the hard segment, the micro-phase separation, and the mechanical properties [22,23]. The polyurethane hard segment content was high in this test. Rigid, hard segments are strongly polar and exist in the structure as physical cross-linking points. The physical cross-linking between the hard segment molecules produces hydrogen bonds. Even though the hydrogen bonding force is smaller than the intermolecular bonding force, a large number of hydrogen bonds can make the intermolecular force stronger, thereby increasing the strength and elastic modulus of the PUC. When the temperature is low, the close arrangement of polar hard segments promotes the formation of a large number of hydrogen bonds; moreover, the degree of hydrogen bonding is high, and the degree of ordered hydrogen bonds is high. When the temperature increases, the polar segments receive energy and become active, resulting in a large number of hydrogen bonds disappearing, a low degree of hydrogen bonding, and a low degree of ordered hydrogen bonds [24,25]. On the other hand, a higher relative content of hard segments and a high degree of ordered hydrogen bonds will lead to a high degree of crystallization of the hard segments. For crystallizable polymers, the degree of crystallization can significantly improve the mechanical properties of PUC [26,27]. The change in temperature causes changes in the number of hydrogen bonds and the degree of crystallization, leading to the change in mechanical properties, so that the flexural strength of PUC shows the law of decreasing flexural strength with increasing temperature.

The fibers selected in this work had basically the same appearance under SEM observation. Next, this research selected PF-PUC specimens for analysis. It can be seen from the comparison of the failure planes in Figure 7 that at −40 °C, the PVA fibers are mainly broken; at +40 °C, most of the fibers are pulled out, and the interface debonding and frictional slippage between the fibers and the matrix occur. This shows that, with the increase in temperature, the bonding force between the fiber and the matrix also gradually decreases, which proves the temperature sensitivity of PUC.

### 3.5. Prediction Model Based on BP Neural Network

It can be seen from the above test results that the relationship between the flexural strength of PUC and the volume ratio of each type of fiber and the temperature is not all linear, so the BP neural network is introduced here. The BP neural network has been widely used in various disciplines and achieved remarkable results. It is an adaptive and self-learning algorithm for solving nonlinear problems.

This research took the experimental data of 30 groups of PF-PUC and CF-PUC and 20 groups of SF-PUC as the sample data for the BP neural network. This research then randomly selected four to five groups as test samples for each, and used the others as training samples, taking each fiber volume ratio and temperature as the input layer, and the 28 d compressive strength of PUC as the output layer. The number of neurons in the hidden layer was calculated and finally determined by using the empirical formula (Formula (2)), where *m* = 2 and *n* = 1. After repeated debugging, the overall performance was better when *L* = 10.
(2)$$L=\sqrt{m+n}+a$$
where *L* is the number of hidden neurons, *M* is the number of neurons in the input layer, *n* is the number of neurons in the output layer, and *a* is a constant, desirable (1, 10).

This model adopts a four-layer network structure, as shown in Figure 8. The input layer is one layer, the hidden layer is two layers, and the output layer is one layer. The number of nodes in the input layer is two, and the number of nodes in the output layer is one. The number of learning steps is 10,000 steps, and the learning target is set to 0.00001.

The mean relative error between the predicted value and the measured value obtained by the training of the BP neural network model is shown in Table 4, in which the mean relative error of SF-PUC is slightly larger, reaching 9%. Figure 9 shows the reliability test between the flexural strength predicted by the BP neural network model and the test results. The fitting degree of the scatter points of PF-PUC and CF-PUC to the straight line y = x reaches 0.99, indicating high reliability. However, SF-PUC has a fit of 0.71 due to too few training samples. It can be seen that the prediction effect of the BP neural network is suitable and can adequately guide the flexural strength prediction and mix ratio design of PF-PUC, CF-PUC, and SF-PUC.

## 4. Conclusions

In this paper, the composition and preparation processes of fiber-reinforced polyurethane cement (PUC) composites were firstly introduced, and PVA fibers, carbon fibers, and steel fibers were selected for PUC reinforcement experiments. Bending tests at different temperatures (−40 °C, −20 °C, 0 °C, 20 °C, and 40 °C) were carried out to obtain the optimum volume-mixing ratio of each type of fiber. The mechanism of fiber toughening and the reason of temperature sensitivity of PUC were analyzed. A prediction model of the flexural strength of fiber-reinforced PUC was proposed. The conclusions are as follows:The three fibers can significantly improve the toughness of PUC. With the increase in three kinds of fiber reinforcement, the flexural strength of the composite at each temperature first increased and then decreased. The incorporation of excess fiber can affect workability and adversely affect the mechanical properties. The best volume-doping ratios of PVA fiber, carbon fiber, and steel fiber were determined by the tests as 0.3%, 0.04%, and 1.0%, respectively. This suggests that the incorporation of fiber to enhance PUC is a feasible approach.PVA fiber has a more obvious increase in the flexural strength of PUC, up to 58.5 MPa at room temperature. Under optimal doping, the ultimate micro-strain of PF-PUC and CF-PUC specimens is smaller, and the specimen stiffness is larger; moreover, the ultimate micro-strain of SF-PUC specimens is larger, and the toughening effect is more obvious. In actual engineering, it is recommended to use PVA fibers for reinforcement scenarios that seek to increase bearing capacity, and steel fibers for reinforcement scenarios that seek to increase deformation capacity.Temperature has a great influence on the flexural strength of PUC, and the flexural strength decreases by an average of 30% when the temperature is between −40 °C and +40 °C. The flexural strength of PUC with and without three kinds of fibers is inversely related to the temperature. This shows that the temperature sensitivity of PUC flexural strength is mainly determined by the polyurethane material itself, and the type and dosing of fibers have less influence on it. The temperature factor should be taken into account when applying PUC to the reinforcement of old bridges to avoid hazards.The established two-factor (fiber content and temperature) BP neural network flexural strength prediction model can predict the flexural strength of PUC under different mixing ratios with high accuracy. The predictive model reduces the cost of the test process to improve efficiency.

## Figures and Tables

**Figure 1 materials-15-04318-f001:**
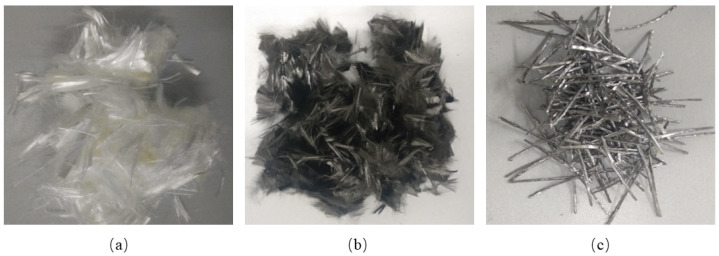
Fiber appearance. (**a**) PVA Fiber; (**b**) Carbon Fiber; (**c**) Steel Fiber.

**Figure 2 materials-15-04318-f002:**
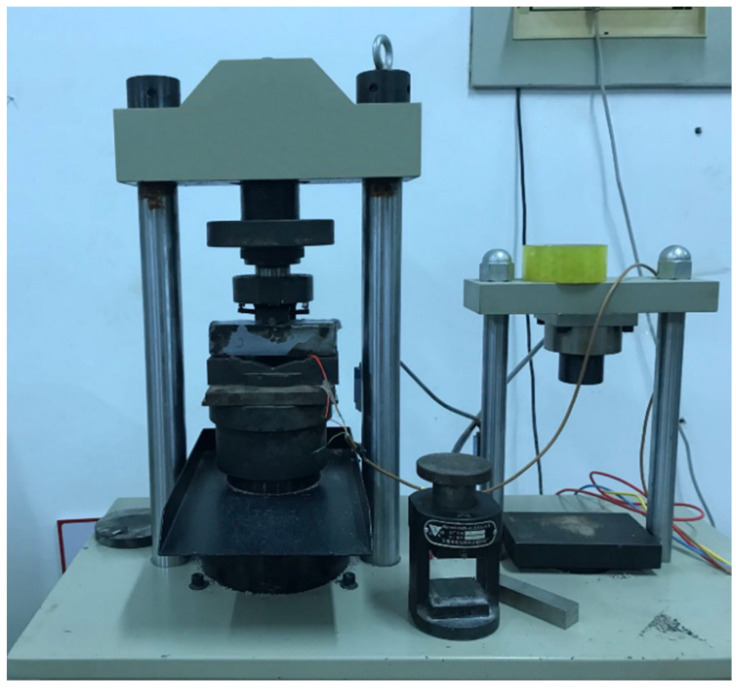
Bending test.

**Figure 3 materials-15-04318-f003:**
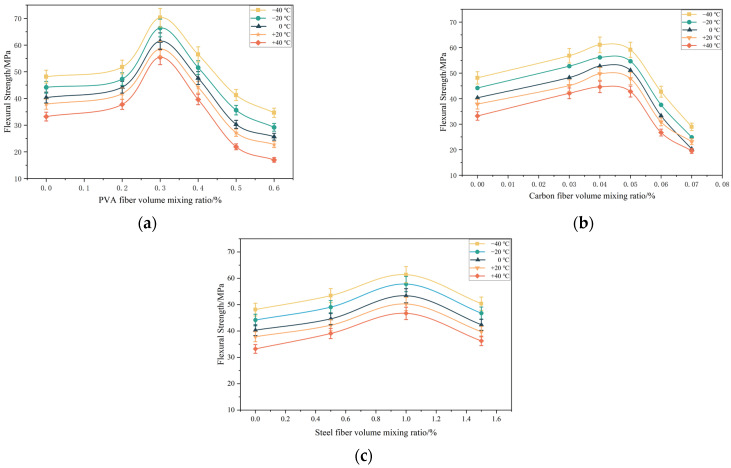
Variation curves of flexural strength with various fiber volume ratios. (**a**) PF-PUC; (**b**) CF-PUC; (**c**) SF-PUC.

**Figure 4 materials-15-04318-f004:**
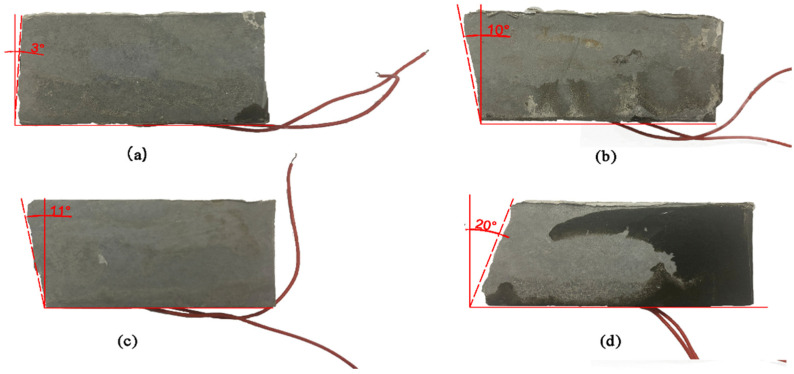
Typical fracture morphology of each group. (**a**) Control; (**b**) PF-PUC; (**c**) CF-PUC; (**d**) SF-PUC.

**Figure 5 materials-15-04318-f005:**
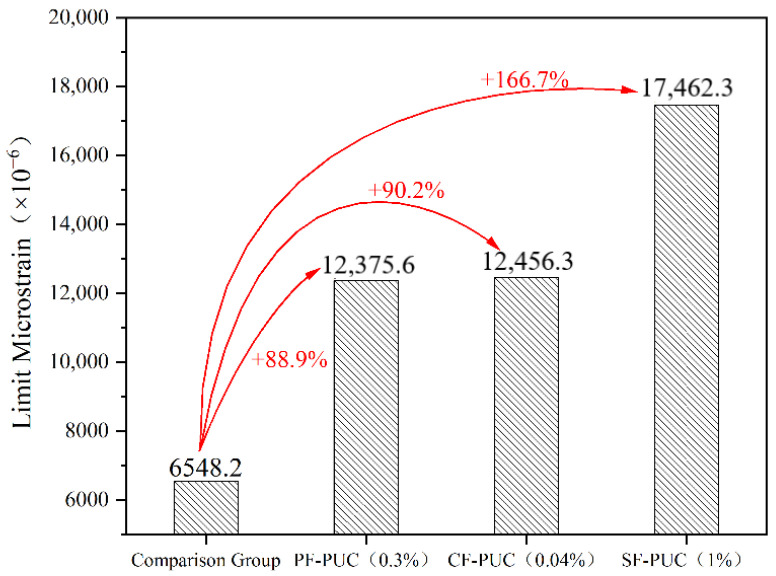
Ultimate micro-strain at optimal volume doping ratio (normal temperature 20 °C).

**Figure 6 materials-15-04318-f006:**
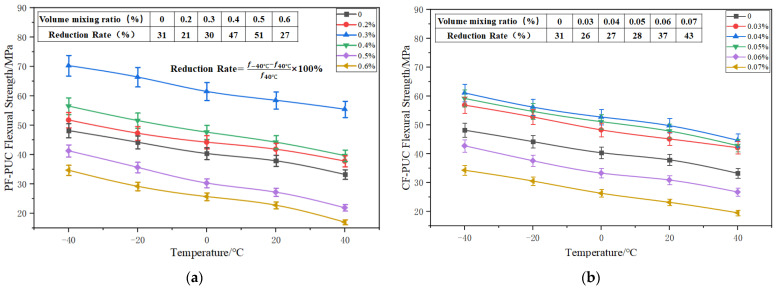
Flexural strength–temperature curves of different fiber volume ratios. (**a**) PF-PUC; (**b**) CF-PUC; (**c**) SF-PUC.

**Figure 7 materials-15-04318-f007:**
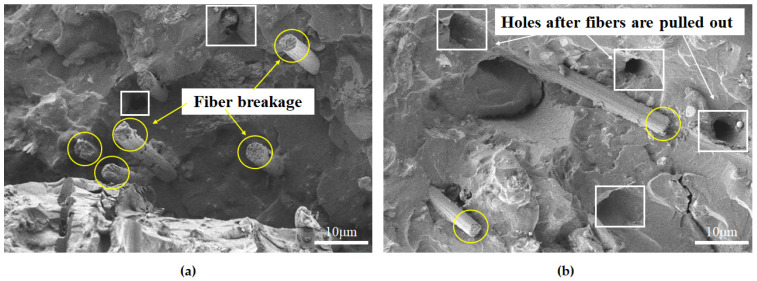
SEM image of typical failure section of PF-PUC specimen. (**a**) −40 °C; (**b**) +40 °C.

**Figure 8 materials-15-04318-f008:**
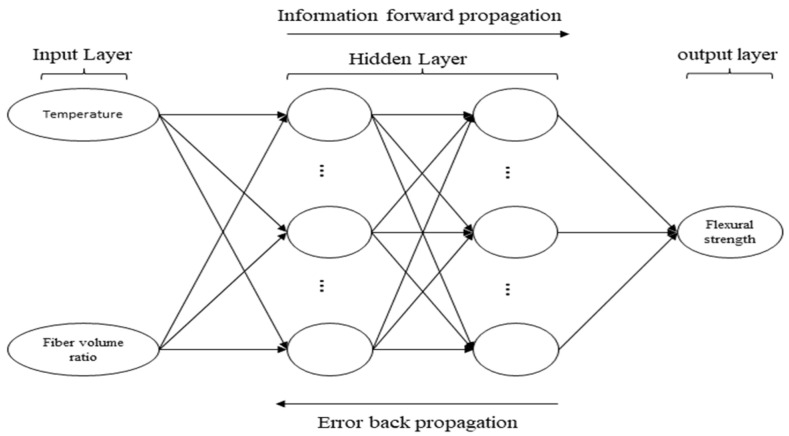
Topological structure of BP neural network.

**Figure 9 materials-15-04318-f009:**
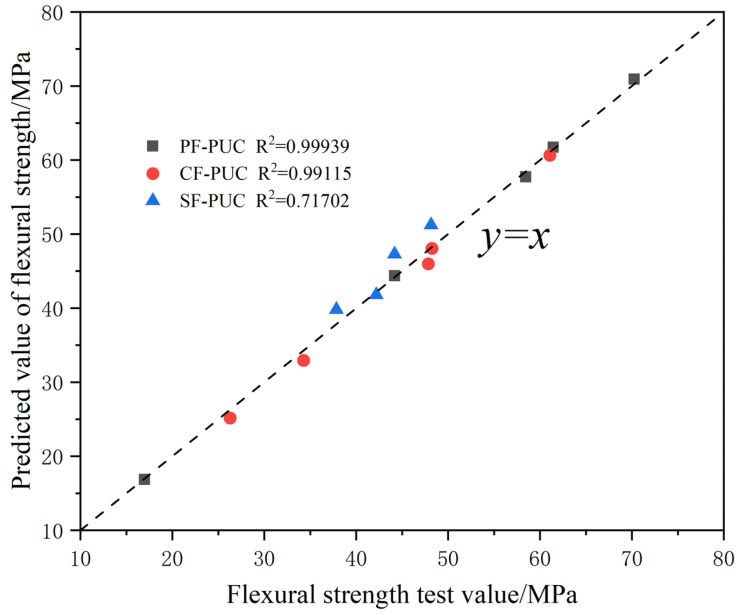
The relationship between the test value and predicted value of the flexural strength of the tested samples.

**Table 1 materials-15-04318-t001:** The mass ratio of each component of PUC.

Ingredients	Cement	Isocyanate	Modified Polyether	Defoaming Agent
Percentage (%)	33	33	33	1

**Table 2 materials-15-04318-t002:** Fiber specifications.

Fiber	Length (mm)	Diameter (μm)	Density (kg/m^3^)	Elastic Modulus (Gpa)	Ultimate Elongation (%)
PVA Fiber	15	8	1290	35	6
Carbon Fiber	15	7	1750	260	1
Steel Fiber	15	200	7850	200	4

**Table 3 materials-15-04318-t003:** The mix ratio of each group of test pieces.

	Numbering	Fiber Volume Ratio (%)	Fiber Weight (kg/m^3^)	Flexural Strength (Mpa)				
				−40 °C	−20 °C	0 °C	+20 °C	+40 °C
Control	/	/	/	48.15	44.18	40.33	37.85	33.21
PVA-fiber-reinforced PUC (PF-PUC)	A1	0.2%	2.58	49.76	45.26	40.21	37.81	32.76
	A2	0.3%	3.87	70.26	66.39	61.47	58.47	55.43
	A3	0.4%	5.16	56.48	51.57	47.62	44.21	39.62
	A4	0.5%	6.45	41.25	35.62	30.29	27.18	21.87
	A5	0.6%	7.74	34.65	29.18	25.67	22.74	16.98
Carbon-fiber-reinforced PUC (CF-PUC)	B1	0.03%	0.53	56.82	52.73	48.25	45.12	42.09
	B2	0.04%	0.70	61.07	56.15	52.72	49.73	44.62
	B3	0.05%	0.88	59.15	54.66	51.09	47.85	42.78
	B4	0.06%	1.05	42.73	37.57	33.28	30.87	26.72
	B5	0.07%	1.23	34.28	30.51	26.29	23.14	19.52
Steel fiber reinforced PUC (SF-PUC)	C1	0.5%	39.25	53.43	49.07	44.62	42.18	39.06
	C2	1%	78.50	61.39	57.82	53.39	50.26	46.71
	C3	1.5%	117.75	50.39	46.77	42.39	39.73	36.25
	C4	2%	157.00	/	/	/	/	/
	C5	2.5%	196.25	/	/	/	/	/

Notes: Groups C4 and C5 failed to prepare test blocks due to excessive steel fiber volume ratio.

**Table 4 materials-15-04318-t004:** Relative error mean of PF-PUC, CF-PUC, and SF-PUC models.

	PF-PUC	CF-PUC	SF-PUC
Relative error mean	2.06%	3%	9%

## Data Availability

Not applicable.

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
