# Peer review of "Study on Toughening and Temperature Sensitivity of Polyurethane Cement (PUC)"

_materials, 2022, doi:10.3390/ma15124318_

Round 1

Reviewer 1 Report

Use of abbreviations in the abstract can be avoided.

When citing research work, please also mention the year of publication. Citations can be brought in line with standard requirements.

The grammar tense of the paper has to be consistent. Language at very few instances is objectionable.

Is it necessary to develop a material that would behave appropriately in both cold and hot regions?

A few of the abbreviations have not been introduced in the main manuscript.

The literature review section of the paper is weak. There are no references to other literature in the results and discussion section as well.

Why is not the type of fibre (and its associated properties) not one of the input layers for the model?

What are liquid A and liquid B?

How many samples were made for each proportion?

Check the SI representation of units in Table 2.

Table 3 has a typographical error.

The sentences are unnecessarily lengthy in section 2.2. Sentence formation is complicated.

Representation of information in tables is not satisfactory.

What is dry maintenance?

Side length and fulcrums are not the right terms to be used for intended parameters. They are width and supports respectively.

Fig 2 title is absurd. It is not a diagram, it’s a set up.

The temperature variable is just temperature and not ‘ambient temperature’.

The increase in flexural capacities mentioned section 3.1 is with respect to?

What is working stage?

On what basis is the role of fibres with regard to the toughening mechanism been presented? How are the authors being able to say what is been discussed in section 3.2?

Do all the specimens of a mix fail the same way as depicted in Fig. 4?

Why weren’t the other categories of specimens tested under SEM?

Reviewer 2 Report

The authors intend to investigate the influences of various fibers such as PVA, carbon and steel to enhance the toughness of conventional polyurethene cement composite. They have measured the flexural strength and developed a model associated to this system. The authors shall consider the following comments and suggestions. Major revision is recommended.

1. The abstract is very ambiguous and confusing. The problem statement / hypothesis is missing and to be included. Several acronyms are not defined properly. The novelty of the work should be presented. 

2. The introduction is not that appealing to the present investigation. It is rather collection of simple results. Gap analysis and shortcomings in fiber reinforced PUC should be consolidated and correlated. This section requires major revision

3. Chemical composition of liquid A and liquid B to be included. What grade of cement is used ?

4. What are the IS codes followed in the mix ratio and test methods ? Without the standard codes, it would be difficult for the readers.

6. The criteria for the choice of fiber should be included

7. Presentation of figure 3 should be improved. Decrement in flexural strength values should be justified. Include the error bars for all the flexural strength values

8. Similar to figure 3, error bars are required for figure 6. On what basis the temperature range was chosen ?

9. Figure quality to be improved at the time of revised submission.

Reviewer 3 Report

The theme of the article is quite original having under consideration the main objective.

However, there are aspects of research methodology that the authors need to make explicit. Therefore:

- In the chapter 2.2 states that "Six groups of test blocks with different doping amounts (0, 0.2%, 0.3%, 0.4%, 0.5%, and 0.6%) were prepared for the flexural test of polyurethane cement (PPUC) doped with PVA fibers, and five test blocks were prepared for each group "and something else is presented in Table 3 (" Fiber Type "column). As shown in Table 3 in the "Fiber Type" column, a typographical error has occurred which needs to be corrected.

- It is also necessary to justify why the values ​​of the volumetric ratios of the fibers were so low and if the values ​​declared in the column "Fiber Volume Ratio" in table 3 are correct, given that it is not explained how they are calculated.

- In the chapter 3.4, in phrase "It can be inferred from Section 2.3 that the flexural strengths of the four groups of specimens all show a law of decreasing with the increase in temperature when the ambient temperature is −40 ° C – 40 ° C" must be reformulated for conformity as "−40 ° C - +40 ° C".

- The phenomenon of expansion-contraction that occurs in the case of temperature variations is known. Therefore, information should be provided on the influence of this phenomenon on the values ​​calculated using formula (1), if the actual dimensions of the specimens were taken into account in the calculation.

- A table with the values ​​obtained in the experimental tests from which the values ​​in the graphs shown in figure 3 were calculated and how many specimens were tested for each case considering that different values ​​may appear due to the inhomogeneity of the mixture.

- The authors have to explain why at the increased values ​​of the volumetric ratios of fibers in the graphs in figure 3, the flexural strength decreases.

- It is recommended to use the sign "+" for positive temperatures.

- It is recommended to use the phrase "fiber reinforcement" instead of "fiber doping", the term "reinforcement" being frequently used in these cases in the technical literature.

- Figure 6 shows the "Volume mixing ratio" which is not explained.

- Conclusions must be reformulated clearly and in relation with the added explanations requested before.

Reviewer 4 Report

The language of the manuscript is very poor. It should be definitely improved.
Avoid using pronoun such as "we"
Important results should be given in abstract
It should be noted that increasing fiber ratio may adversely affect the workability (doi.org/10.1016/j.jobe.2020.102119)
More detail should be provided for Neural Network analysis 
More discussions should be provided for Figure 9.
It should better write some summary in conclusion before writing important conclusions
Some recommendations and future studies can be added to concluision

(1)Conclusion section summarizes the outcomes of the study; however, no recommendation was provided. Therefore, conclusion section should be improved.

(2)The presentation of the paper is good. The results are supported with graphs and SEM.

(3) The methodolgy is presented in detail.

(4) Novality is not clear. The author should emphasize the novality of this study.

Reviewer 5 Report

1.Need to clearify abd add the novel results "Given the results, we established a prediction model of the flexural strength of fiber-reinforced composites based on a two-factor BP neural network."

2.Rewrite "According to the existing experimental data, we designed a BP neural network flexural strength prediction model with the fiber volume ratio and ambient temperature as the input layer, and verified that the model can predict well with suitable accuracy. It provides a reference for the future use of PUC for bridge reinforcement designs in different regions."

3.Need to explain more "The test block number is shown in Table 3."

4.Recheck "The reason for the increase in flexural strength is that when the test block is prepared, the fibers are randomly distributed inside the material and form a network with PUC, which makes the network tightly connected, and the integrity is greatly improved. Greater pulling force is required to break the fibers or pull them out of the PUC."

5.Rewrite "When micro-cracks appear in the PUC matrix, the three types of fibers with high ductility straddle the microcracks to relieve sharp stress concentration and transfer the stress to the surrounding uncracked part of the matrix, and the stress distribution tends to be uniform. When resisting flexural failure, all three fibers were pulled out, slipped or fractured relative to the PUC matrix, absorbed a large amount of failure energy, and significantly improved the toughness of the composite material."

6.Revise "The polyurethane hard segment content in this experiment is high, the rigid hard segment has strong polarity and exists in the structure as a physical crosslinking point, and the physical crosslinking between the hard segment molecules generates hydrogen bonds. Although the force of hydrogen bonds is smaller than the bonding force, the number of hydrogen bonds is greater, and the physical cross-linking of hydrogen bonds makes the intermolecular force stronger, which will improve the mechanical properties such as the strength and elastic modulus of PUC."

7.Specify "The temperature sensitivity of PUC flexural strength is mainly determined by the polyurethane material itself, and the type and dosing of fibers have less influence on it."

Round 2

Reviewer 2 Report

The manuscript may please be accepted

Reviewer 4 Report

The authors completed the reviewer requests.

The paper can be published in this current form.

Reviewer 5 Report

It can be published.